# Post-recovery COVID-19 and incident heart failure in the National COVID Cohort Collaborative (N3C) study

Husam M. Salah[1], Marat Fudim [2,3✉], Shawn T. O'Neil[4], Amin Manna [5], Christopher G. Chute [6] & Melissa C. Caughey [7]

Cardiac involvement has been noted in COVID-19 infection. However, the relationship between post-recovery COVID-19 and development of de novo heart failure has not been investigated in a large, nationally representative population. We examined post-recovery outcomes of 587,330 patients hospitalized in the United States (257,075 with COVID-19 and 330,255 without), using data from the National COVID Cohort Collaborative study. Patients hospitalized with COVID-19 were older (51 vs. 46 years), more often male (49% vs. 42%), and less often White (61% vs. 69%). Over a median follow up of 367 days, 10,979 incident heart failure events occurred. After adjustments, COVID-19 hospitalization was associated with a 45% higher hazard of incident heart failure (hazard ratio = 1.45; 95% confidence interval: 1.39–1.51), with more pronounced associations among patients who were younger (*P*-interaction = 0.003), White (*P*-interaction = 0.005), or who had established cardiovascular disease (*P*-interaction = 0.005). In conclusion, COVID-19 hospitalization is associated with increased risk of incident heart failure.

[1] Department of Medicine, University of Arkansas for Medical Sciences, Little Rock, AR, USA. [2] Duke Clinical Research Institute, Duke University School of Medicine, Durham, NC, USA. [3] Division of Cardiology, Duke University School of Medicine, Durham, NC, USA. [4] Center for Health AI, University of Colorado Anschutz Medical Campus, Aurora, CO, USA. [5] Palantir Technologies, Denver, CO, USA. [6] Schools of Medicine, Public Health, and Nursing, Johns Hopkins University, Baltimore, MD, USA. [7] Joint Department of Biomedical Engineering, University of North Carolina and North Carolina State University, Chapel Hill, NC, USA. ✉email: marat.fudim@gmail.com

The coronavirus disease 2019 (COVID-19), caused by severe acute respiratory syndrome coronavirus 2 (SARS-CoV-2), has resulted in unprecedented morbidity and mortality with several long-term complications, including cardiac and respiratory disorders and an increased risk of premature death, among those who recover[1,2]. Despite the absence of significant lung damage in many patients discharged from COVID-19 hospitalization, post-recovery dyspnea and fatigue have been reported, with almost 87% of patients experiencing persistent, post-discharge symptoms[3]. The 'long-hauler' phenomenon has been described in patients suffering from chronic symptoms following recovery from COVID-19 infection[4]. However, despite its important clinical implications, there is limited understanding of the pathophysiology and etiology driving these symptoms.

Because inflammation is a central pathophysiological mechanism shared by both COVID-19 and heart failure (HF)[5], an association between COVID-19 and incident HF may exist. In support of this, among 3080 consecutive patients presenting to the emergency department with COVID-19 infection, 60 without prior history of HF were subsequently admitted for acute HF[6]. Further, an analysis from the US Department of Veterans Affairs (VA) national healthcare system showed that military veterans who recovered from COVID-19 were at increased risk of incident cardiovascular diseases, including heart failure[7]. This study demonstrates a significant burden of incident heart failure among COVID-19 survivors; however, the demographic composition, which consisted of veterans who were mostly older males, limits the generalizability of this association to the general population[7]. Herein, we examine the association between COVID-19 recovery and incident HF in a large-scale, population-based, nationally representative study.

## Results

At the time of our analysis, the N3C enclave included electronic health records (EHRs) for inpatient and outpatient encounters for over 12 million unique patients. We limited our study population to patients with a hospitalization occurring during the March 1, 2020–March 31, 2022 interval. A total of 355,673 unique patients with a first-occurring COVID-19 diagnosis date during inpatient hospitalization were identified. Of these, 257,075 survived to discharge and had no prior history of HF. During the same time interval, a total of 2,560,320 unique patients hospitalized without COVID-19 diagnosis were identified. To speed computer processing times, a random sample (15% of the total) was taken, amounting to 384,048 unique patients without COVID-19 infection. Of these, 330,255 survived to discharge and had no prior history of HF. Our total study population included 587,330 unique patients, with the selection flowchart shown in Fig. 1.

Patients hospitalized with COVID-19 were generally older (51 vs. 46 years), more often male (49% vs. 42%), and less often White (61% vs. 69%). The prevalence of diabetes was notably higher among patients hospitalized with COVID-19 (24% vs. 16%). History of hypertension (42% vs. 38%) and chronic kidney disease (11% vs. 8%) were also slightly more prevalent; however, the prevalence of obesity, coronary artery disease and chronic lung disease were largely comparable between the two groups (Table 1). Patients with COVID-19 hospitalization were similarly likely to be managed by an angiotensin converting enzyme inhibitor (ACEi), angiotensin II receptor blockers (ARBs), or statins, but less likely to be managed by beta blockers (22% vs. 26%).

Over a median follow up of 367 (25th–75th percentiles: 190–536) days, 10,979 incident HF events and 17,641 deaths occurred. Patients discharged from COVID-19 hospitalizations had a higher cumulative incidence of post-discharge HF (2.3% vs. 1.5%), a higher mortality rate (3.3% vs. 2.6%) and a higher

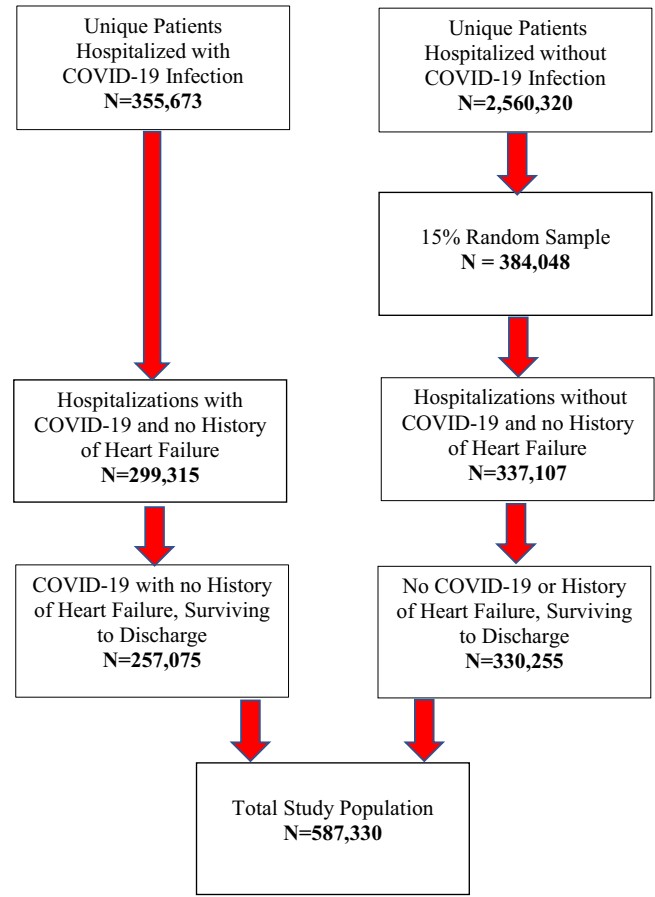

**Fig. 1 Study population flowchart.** The total study population included 587,330 unique patients as in the following selection flowchart.

cumulative incidence of composite death or incident HF (5.2% vs. 4.0%) than patients hospitalized without COVID-19 (Table 2). The crude incidence rate of HF was also higher for patients discharged from COVID-19 hospitalization (2.55 events per 100 person-years [p.y.] vs. 1.44 events per 100 p.y.). The median follow up to incident HF was shorter for patients discharged from COVID-19 hospitalization (69 vs. 84 days). However, a similar proportion of incident HF cases occurred within 30 days of discharge, when comparing patients with COVID-19 hospitalization (1914/5879, 33%) to those without COVID-19 hospitalization (1506/5100, 30%).

Biomarkers of cardiac injury were not commonly assayed at the index hospitalization. However, a higher percentage of patients with COVID-19 hospitalization had measured natriuretic peptides (23% vs. 6%) or cardiac troponins (21% vs. 8%), compared to patients without COVID-19 hospitalization. Median troponin levels were similar between the two groups, while median levels for natriuretic peptides were slightly higher in patients without COVID-19 hospitalization. Among the cases with subsequent incident HF, elevated natriuretic peptides were noted at the index hospitalization in 19% and 12%, respectively, for patients with and without COVID-19 hospitalization (Table 3).

Without adjustments, post-recovery COVID-19 was associated with a 69% higher hazard of post-discharge incident HF (HR = 1.69; 95% CI: 1.63–1.76), which attenuated to a 45% higher hazard after accounting for age, race/ethnicity, sex, heart failure risk factors, and cardiovascular medications (HR = 1.45; 95% CI: 1.39–1.51). (Table 4). When excluding patients with evidence of

elevated natriuretic peptides at the index hospitalization, the fully adjusted model yielded a 51% higher hazard of incident HF for patients with vs. without post-recovery COVID-19 (HR = 1.51;

**Table 1 Baseline demographic and clinical characteristics of patients hospitalized with and without COVID-19.**

| Characteristic | COVID-19 Hospitalization N = 257,075 | No COVID-19 Hospitalization N = 330,255 | P-value* |
|---|---|---|---|
| Demographics | | | |
| Age† (years, mean ± S.D.) | 51 ± 22 | 46 ± 23 | <0.0001 |
| Male† | 125,684 (49%) | 140,147 (42%) | <0.0001 |
| Race/Ethnicity† | | | <0.0001 |
| White | 144,990 (61%) | 214,616 (69%) | |
| Black | 56,910 (24%) | 58,682 (19%) | |
| Asian | 7391 (3%) | 10,266 (3%) | |
| Other‡ | 30,293 (13%) | 26,798 (9%) | |
| Heart failure risk factors | | | |
| Hypertension | 106,810 (42%) | 126,911 (38%) | <0.0001 |
| Obesity | 56,640 (22%) | 65,507 (20%) | <0.0001 |
| Coronary artery disease | 21,518 (8%) | 28,317 (9%) | 0.005 |
| Diabetes | 60,682 (24%) | 54,371 (16%) | <0.0001 |
| Chronic kidney disease | 28,476 (11%) | 25,440 (8%) | <0.0001 |
| Chronic lung disease | 17,548 (7%) | 19,945 (6%) | <0.0001 |
| Cardiovascular medications | | | |
| Angiotensin converting enzyme inhibitor | 33,335 (13%) | 39,837 (12%) | <0.0001 |
| Angiotensin II receptor blocker | 19,749 (8%) | 22,098 (7%) | <0.0001 |
| Beta Blocker | 56,444 (22%) | 84,302 (26%) | <0.0001 |
| Statin | 60,936 (24%) | 69,892 (21%) | <0.0001 |

The National COVID Cohort Collaborative Study
*S.D.* standard deviation
*Groups compared by 2-sample t-tests or Pearson chi-square tests.
†Age missing for 14,569 (2%), sex missing for 156 (0.03%), race/ethnicity missing for 37,384 (6%).
‡Other signifies multi-race, other race, or Hispanic ethnicity.

95% CI: 1.44–1.57). Higher adjusted hazards of incident heart failure were noted in patients ≤65 years (HR = 1.53; 95% CI: 1.44–1.64) compared to those >65 years (HR = 1.38; 95% CI: 1.31–1.45); P-interaction = 0.003 (Fig. 2). Adjusted hazards of incident HF were also higher for White patients (HR = 1.49; 95% CI: 1.42–1.57) compared to non-White or Hispanic patients (HR = 1.36; 95% CI: 1.27–1.46), P-interaction = 0.005; and for patients taking cardiovascular medications (HR = 1.48; 95% CI: 1.42–1.55) compared to patients who were not managed by cardiovascular medications (HR = 1.30; 95% CI: 1.19–1.42), P-interaction = 0.005. The unadjusted hazard ratios of incident HF associated with post-recovery COVID-19 hospitalization are shown in Table 4 and Supplementary Table 1.

## Discussion

In this large-scale, population-based, nationally representative sample of 587,330 hospitalized patients, we observed a higher incidence of post-discharge HF in patients with COVID-19 hospitalization. The heightened risk of post-recovery incident heart failure persisted, even after adjustment for demographics (age, sex, race/ethnicity), HF risk factors (hypertension, obesity, coronary artery disease, diabetes, chronic kidney disease, and chronic lung disease) and established cardiovascular disease indicated by cardiovascular medication use (ACEi, ARBs, beta blockers, or statins). The risk of post-discharge incident HF was more pronounced among patients who were young (i.e., <65 years old), White, and those with established cardiovascular disease.

Consistent with the results of a recent analysis from the US Veterans Affairs national healthcare system[7], we show that COVID-19 is associated with an increased hazard of post-recovery incident heart failure. Our analysis from the N3C study distinguishes itself as the first large-scale, nationally representative analysis that overcomes the limited demographic composition of the US Veterans Affairs hospitals (i.e., military veterans who were mostly men)[7].

The pathophysiological mechanisms that underlie the association between COVID-19 and incident HF are likely complex and multifactorial. COVID-19 is associated with endothelial activation and dysfunction and a prothrombotic state[8,9], which can result in macro- and microvascular coronary thrombosis with subsequent cardiac dysfunction. Further, an accumulating body

**Table 2 Summary of post-hospitalization follow up and events among patients with and without COVID-19 hospitalization.**

| Descriptor | Covid-19 Hospitalization | No Covid Hospitalization |
|---|---|---|
| Entire sample | | |
| Number of patients | 257,075 | 330,255 |
| Total follow up (person-years)* | 230,296 | 353,464 |
| Median (Q1-Q3) follow up (days) | 331 (149–511) | 405 (232–562) |
| Heart failure | | |
| Number of incident heart failure cases | 5879 | 5100 |
| Heart failure cumulative incidence | 2.3% | 1.5% |
| Median (Q1-Q3) follow up to heart failure event (days) | 69 (20–188) | 84 (23–213) |
| Heart failure incidence rate | 2.55/100 p.y. | 1.44/100 p.y. |
| Deaths | | |
| Number of deaths | 8524 | 9117 |
| Cumulative mortality | 3.3% | 2.6% |
| Median (Q1-Q3) follow up to death (days) | 31 (10–115) | 66 (21–178) |
| Composite events | | |
| Number of heart failure or death events† | 13,255 | 13,375 |
| Composite events cumulative incidence | 5.2% | 4.0% |
| Composite events incidence rate | 5.76/100 p.y. | 3.78/100 p.y. |

The National COVID Cohort Collaborative Study.
*Q1* quartile 1, *Q3* quartile 3, *p.y.* person years.
*Follow up time accrued since hospital discharge until occurrence of heart failure event, death, or end of surveillance (whichever first).
†Composite events based on number of unique patients experiencing incident heart failure, death, or both.

**Table 3 Cardiac biomarkers assayed at index hospitalization for patients with and without COVID-19 infection.**

| Biomarker laboratories | COVID-19 Hospitalization (N = 257,075) | | No COVID-19 Hospitalization (N = 330,255) | |
|---|---|---|---|---|
| | Number tested | Median (25%–75%) | Number tested | Median (25%–75%) |
| Natriuretic peptides | | | | |
| BNP (pg/mL) | 29,551 (11%) | 50 (21-133) | 10,580 (3%) | 74 (30-203) |
| NT-proBNP (pg/mL) | 34,251 (13%) | 185 (59-696) | 9442 (3%) | 286 (83-996) |
| Cardiac troponins | | | | |
| Troponin I (ng/mL) | 30,496 (12%) | 0.02 (0.01-0.05) | 16,058 (5%) | 0.02 (0.01-0.07) |
| Troponin T (ng/mL) | 22,105 (9%) | 0.01 (0.01-0.03) | 10,517 (3%) | 0.01 (0.01-0.03) |
| | Subsequent Heart Failure (N = 5879) | No Subsequent Heart Failure (N = 251,196) | Subsequent Heart Failure (N = 5100) | No Subsequent Heart Failure (N = 325,155) |
| Elevated Natriuretic Peptide* | 1103 (19%) | 23,302 (9%) | 598 (12%) | 8588 (3%) |

The National COVID Cohort Collaborative Study.
BNP B-type natriuretic peptide, NT-proBNP N-terminal prohormone brain natriuretic peptide.
*Natriuretic peptides considered elevated if BNP >100 pg/mL or NT-proBNP >300 pg/mL.

**Table 4 Multivariable Cox regression models analyzing hazard ratios of post-discharge incident heart failure among patients with and without COVID-19 hospitalization.**

| Model* | HR (95% CI) |
|---|---|
| Incident heart failure | |
| Crude | 1.69 (1.63-1.76) |
| Demographics (age, race/ethnicity, sex) | 1.37 (1.32-1.42) |
| Demographics and risk factors† | 1.40 (1.34-1.45) |
| Demographics, risk factors, and medications‡ | 1.45 (1.39-1.51) |

The National COVID Cohort Collaborative Study.
*Models based on 559,017 unique patients without missing age, race/ethnicity, or sex.
†Risk factors = hypertension, obesity, coronary artery disease, diabetes, chronic kidney disease, and chronic lung disease.
‡Medications = Angiotensin converting enzyme inhibitor, angiotensin II receptor blocker, beta blocker, and statins.

of evidence suggests that SARS-CoV-2 can promote metabolic changes that favor viral survival[10]; such changes can subsequently alter cell phenotype and function and result in sustained inflammation and tissue injury[10]. Cardiac injury and myocardial inflammation, both of which have repeatedly been reported in previous COVID-19 studies, may be central to the pathogenesis of incident HF following COVID-19.

Our analysis suggests a potential role for COVID-19 infection in the development of incident HF. To a varying degree, our findings are supported by previous clinical observations. In a hospital registry of patients recovered from COVID-19, cardiac involvement was evident in 78%, with ongoing myocardial inflammation in 60%, independent of pre-existing medical conditions[11]. However, in competitive athletes with mild or asymptomatic COVID-19 infection, the reported incidence of myocardial inflammation ranged from 0% to 15%[12,13]. These discrepancies in reported inflammation may be related to the study populations or severity of COVID-19 infection. In an analysis of hospitalized patients with COVID-19 in the Mount Sinai Health System, cardiac injury (as measured by an abnormal troponin-I level) was evident in 36% and was significantly associated with mortality[14]. On the other hand, we did not observe differences in mean troponin levels at the index hospitalization, when comparing patients with vs. without COVID-19 infection. This may be partly attributable to the exclusion of existing HF in our study population, as HF is known to elevate troponin levels. We also excluded patients with in-hospital death, which was strongly related to elevated troponin in the Mount Sinai Health System study. Nonetheless, mounting evidence suggests that COVID-19 infection is associated with myocardial inflammation, which may persist even after recovery, as shown by abnormal cardiac magnetic resonance (CMR) imaging findings (e.g., high T1 and/or T2, nonischemic late gadolinium enhancement)[15]. Our population-based investigation extends previous findings relating severe COVID-19 infection to myocardial damage and onset of incident HF, in a large, nationally representative study population.

There are some limitations in this study. First, clinical variables were extracted from the EHR and harmonized across four common data models, but are reflective of real-world practice and may be subject to residual heterogeneity. Second, the median post-discharge follow up time was approximately 1 year, and we were unable to consider longer-term sequelae. Third, we were unable to consider cardiac imaging (e.g., echocardiography, cardiac magnetic resonance) and the majority of HF descriptions were nonspecific, e.g., "congestive heart failure". As such, we were unable to conduct a meaningful analysis of the association between COVID-19 and various HF phenotypes. Fourth, SARS-CoV-2 has undergone frequent mutations with emergence of several variants that have varying degrees of severity[16]; therefore, different SARS-CoV-2 variants may be associated with varying risk of post-recovery incident heart failure. As we do not have data related to SARS-CoV-2 variants, we could not conduct a subgroup analysis to assess the risk of post-recovery incident heart failure among different SARS-CoV-2 variants.

In conclusion, COVID-19 infection appears to be associated with an increased risk of incident HF. A suspicion of HF should be triggered in patients who experience respiratory or cardiac symptoms following recovery from COVID-19 hospitalization.

## Methods

**The National COVID Cohort Collaborative**. The National COVID Cohort Collaborative (N3C, covid.cd2h.org) represents a partnership between the Clinical and Translational Science Awards (CTSA) Program hubs (60 institutions), the National Center for Advancing Translational Science (NCATS), the Center for Data to Health (CD2H), and the community. As previously described[17], N3C is a secure enclave of observational electronic health records (EHR) harmonized across multiple healthcare systems in the United States. Data domains are programmatically extracted from the EHR and standardized to the Observational Medical Outcomes Partnership (OMOP) common data model. The N3C includes patients with known or suspected COVID-19 infection, and by design, a control group without COVID-19 infection. COVID-19 was phenotyped using diagnosis codes, procedure codes, and laboratory codes, in accordance with guidance from the Centers for Disease Control and Prevention. Historical, pre-pandemic EHR were extracted dating back to January 2018, with ongoing record collection continuing to the present day. All N3C activities were approved by a central Institutional Review Board at Johns Hopkins University (Reliance Protocol IRB00249128).

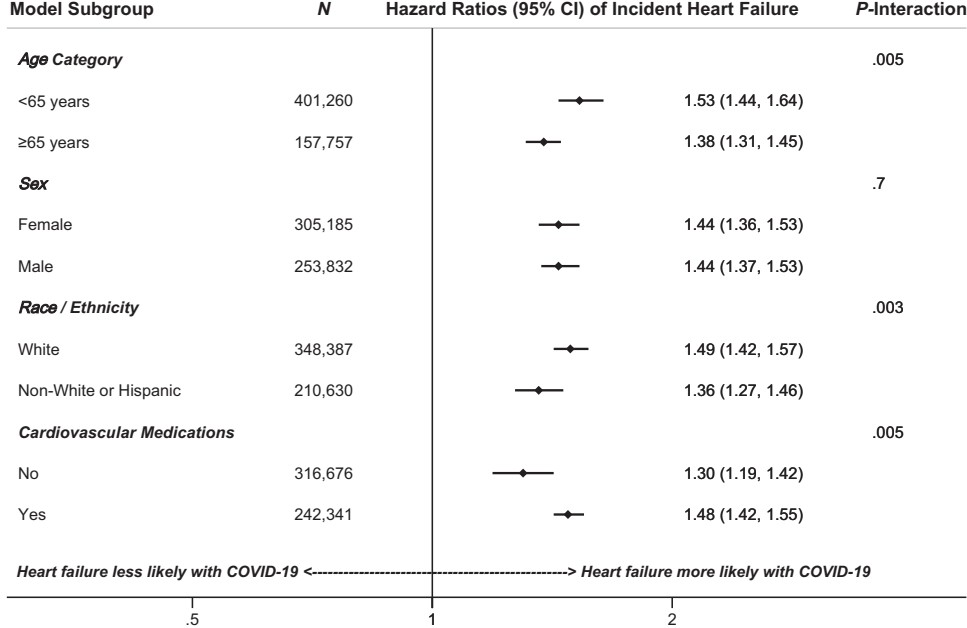

| Model Subgroup | N | Hazard Ratios (95% CI) of Incident Heart Failure | P-Interaction |
|---|---|---|---|
| **Age Category** | | | .005 |
| <65 years | 401,260 | | 1.53 (1.44, 1.64) |
| ≥65 years | 157,757 | | 1.38 (1.31, 1.45) |
| **Sex** | | | .7 |
| Female | 305,185 | | 1.44 (1.36, 1.53) |
| Male | 253,832 | | 1.44 (1.37, 1.53) |
| **Race / Ethnicity** | | | .003 |
| White | 348,387 | | 1.49 (1.42, 1.57) |
| Non-White or Hispanic | 210,630 | | 1.36 (1.27, 1.46) |
| **Cardiovascular Medications** | | | .005 |
| No | 316,676 | | 1.30 (1.19, 1.42) |
| Yes | 242,341 | | 1.48 (1.42, 1.55) |

*Heart failure less likely with COVID-19* <----------------------------------------------------> *Heart failure more likely with COVID-19*

.5　　　　　　　1　　　　　　　2

**Fig. 2 Adjusted hazard ratios of incident heart failure comparing patients hospitalized with versus without COVID-19.** Multivariable Cox regression models based on a total of 559,017 unique patients without missing age, race/ethnicity, or sex. Models adjusted for demographics (age, race/ethnicity, sex), heart failure risk factors (hypertension, obesity, coronary artery disease, diabetes, chronic kidney disease, chronic lung disease) and cardiovascular medications (angiotensin converting enzyme inhibitor, angiotensin II receptor blocker, beta blocker, and statins). Point = hazard ratio, horizontal bars = 95% confidence intervals, vertical bar = null value (hazard ratio = 1.0).

We conducted a secondary analysis of the "Level 2" deidentified N3C dataset, which redacted 17 personal identifiers and date-shifted longitudinal data to protect patient privacy. COVID-19 cases were included in our analysis if the first diagnosis date for COVID-19 occurred during inpatient hospitalization, inclusive of preadmission testing. Non-COVID-19 hospitalizations were identified by absence of a COVID-19 diagnosis, including the time intervals prior to the hospitalization and during the post-discharge follow up. We limited our study population to patients surviving to hospital discharge, and excluded any patients with history of HF documented on or prior to the index hospitalization discharge date. For the purposes of our analysis, index hospitalizations were identified by admissions on or later than March 1, 2020, and end of surveillance was March 31, 2022.

**Data extraction**. Clinically diagnosed conditions were extracted from the N3C enclave if documented in the EHR, using Spark SQL version 3.0.2. Physician-diagnosed HF was used both as an exclusion factor (if documented on or prior to the index hospitalization discharge date), and a post-discharge clinical outcome. Available biomarkers of cardiac injury (B-type natriuretic peptide [BNP], N-terminal prohormone brain natriuretic peptide [NT-proBNP], cardiac troponin I, and cardiac troponin T) were extracted from the index hospitalization, using the first laboratory value in the event of serial testing. For the purposes of this analysis, natriuretic peptides were considered elevated by BNP > 100 pg/mL or NT-proBNP >300 pg/mL. Medical histories (e.g., hypertension, obesity, coronary artery disease, diabetes, chronic kidney disease, and chronic lung disease) were extracted if documented on or prior to the index hospitalization discharge date, as was cardiovascular medication use (angiotensin converting enzyme inhibitor [ACEi], angiotensin II receptor blocker [ARB], beta blockers, and statins). Diagnosed conditions were considered present or absent if noted in the electronic health record. Demographic data was extracted from the index hospitalization, with race/ethnicity categorized into the following groups: White, Black, Asian, or Other, with Other inclusive of muti-races and Hispanic ethnicity. In most cases, the documented race/ethnicity was based on self-report.

**Statistical analysis**. All data management and statistical analyses were conducted and documented in the secure N3C Enclave programming environment. Aggregate counts and summary data were queried using Spark SQL version 3.0.2. Descriptive characteristics were compared by chi-square tests, 2-sample t-tests, or Wilcoxon rank sums tests. Multivariable Cox regression models were analyzed using R version 3.5.1. Hazard ratios of incident HF for patients with post-recovery COVID-19 relative to those without COVID-19 hospitalization were analyzed by Cox regression, adjusted for age, sex, race/ethnicity (White vs. Non-White), hypertension, obesity, coronary artery disease, diabetes, chronic kidney disease, and chronic lung disease, and cardiovascular medication use (ACEi, ARBs, beta blockers, or statins). Post-discharge follow up time accrued until date of incident HF event, death, or end of surveillance, whichever came

first. Potential modification of the association between post-recovery COVID-19 with incident HF was assessed by stratification (age ≤65 vs. 65 years), (females vs. males), (White vs. non-White race/ethnicity), and (cardiovascular medication use vs. no use). Race/ethnic groups were categorized as White vs. non-White to analyze potential modification by racial or ethnic minority status. Significance of the effect modification was assessed by testing by the multiplicative interaction of the stratifying variable with post-recovery COVID-19 status. Because elevated natriuretic peptides at the index hospitalization may indicate existing, undocumented HF, we conducted a sensitivity analysis excluding any patients identified with elevated BNP or NT-proBNP.

**Reporting summary**. Further information on research design is available in the Nature Research Reporting Summary linked to this article.

## Data availability
The data that support the findings of this study may be requested from NCATS (covid.cd2h.org) but restrictions apply. Access to data requires N3C onboarding and following the regulatory and training requirements that all N3C users must abide by.

## Code availability
The analytic code may be accessed at GitHub - mcaughey/n3c_hf

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

## Acknowledgements

The analyses described in this publication were conducted with data or tools accessed through the NCATS N3C Data Enclave https://covid.cd2h.org and supported by NCATS U24 TR002306. This research was possible because of the patients whose information is included within the data and the organizations (see covid.cd2h.org) and scientists who have contributed to the on-going development of this community resource: https://doi.org/10.1093/jamia/ocaa196.

## Author contributions

H.M.S., MF and M.C.C. conceptualized the study. H.M.S and M.C.C wrote the manuscript. M.C.C. performed the statistical analysis. M.F., S.T.O'Neil, A.M., and G.C. interpreted the data and revised the manuscript critically.

## Competing interests

A.M. is an employee of Palantir Technologies, which did not sponsor any of this work. The other authors report no relevant disclosures to the submitted material.
