## [Peer Review File · Nature Communications]

Post-recovery COVID-19 and incident heart failure in the National COVID Cohort Collaborative (N3C) studyReviewers' comments:

Reviewer #1 (Remarks to the Author):

This manuscript by Salah et al. uses a population-level case-control study design to assess incident heart failure in patients after COVID-19 recovery. The authors used the National COVID Cohort Collaborative (N3C) dataset for their analyses. I have the following comments/suggestions for the authors:

Major Comments

1) In their results, the authors state 'A total of 1,735,767 unique patients hospitalized without COVID-19 diagnosis were identified. To speed computer processing times, a random sample (15% of the total) was taken, amounting to 228,170.' With such a large sample size, I don't understand the rationale to randomly sample and then adjust for confounders rather than purposefully select matched controls (matched on age, gender, race, and medical comorbidities). Perhaps dedicated statistical review would be beneficial, but my impression is that matching is a stronger design.

2) In their methods, the authors do not provide sufficient information about how incident HF (the primary outcome) was assessed. In their limitations, the authors state 'We were unable to consider HF biomarkers or cardiac imaging...and the majority of HF documentations were non-specific, e.g. 'congestive heart failure'.' So were only administrative diagnosis codes used, or did they have access to laboratory values (i.e., BNP, NT-proBNP)? This information is critical for readers to understand the internal and external validity of the analyses conducted.

3) Limitations above aside, the authors' analyses as conducted found a significantly higher incidence of HF among patients hospitalized with COVID-19 vs. those hospitalized without COVID-19, even after adjustment for relevant confounders. That is certainly a noteworthy finding, particularly in the large population studied. However, aside from a handful of subgroup analyses, the authors do not include any further in-depth analyses. For example, they don't provide any analyses incorporating troponin I or other markers of cardiac injury. I suspect this may be because they only had access to administrative diagnosis codes and not laboratory data. If they did have access to laboratory data, then I'd like to see some of that data and how it influenced the stated findings. If they didn't have access, then that should be clearly stated in the manuscript.

Minor Comments

1) In their methods, the authors state they excluded 'any patients with history of HF within 2 years prior to the hospitalization.' Why did the authors not choose to exclude patients with ANY history of HF? Why the 2-year cutoff?

Reviewer #2 (Remarks to the Author):

In this manuscript, Salah and cool report the incidence of HF after COVID infection. Using data from the N3C consortium, they examined incidence of HF post-discharge in patients with and without COVID infection concluding that COVID is associated with increased risk of incident HF. The manuscript is well written. However, the novelty is somehow limited as similar data are variably published over the last two years. Furthermore, the amount of data presented and the analysis performed may not be sufficient for a full paper.

In addition there are several major issues limiting the impact of the manuscript.

Major comments:

1. please refrain from using specific dates in the introduction.
2. The introduction may be significantly shortened and resented to improve clarity
3. Please use the correct manuscript formatting as per journal style
4. The discussion is quite superficial. It does not encompass the complexity of the problem and does not refer to the appropriate literature for discussion.
5. The authors should investigate the causes of the increased incidence of HF as well as the time to hospitalization. Was this within 30 days from discharge? Please divide in subgroups accordingly.
6. The lack of clinical characteristics is a critical limitation.

7. Table 1 is minimal. IT should be significantly expanded with characteristics of patients at discharge and, if possible, with those at first HF event
8. The patients with unknown race/ethnicity should be excluded and not counted as "other"
9. Please report the % of missing variables
10. A table with HRs in univariate and multivariate should be reported. However the lack of clinical variables downplays the significance of the analysis as the correction might be underestimated.

Reviewer #1 (Remarks to the Author):

1) In their results, the authors state 'A total of 1,735,767 unique patients hospitalized without COVID-19 diagnosis were identified. To speed computer processing times, a random sample (15% of the total) was taken, amounting to 228,170.' With such a large sample size, I don't understand the rationale to randomly sample and then adjust for confounders rather than purposefully select matched controls (matched on age, gender, race, and medical comorbidities). Perhaps dedicated statistical review would be beneficial, but my impression is that matching is a stronger design.

Thanks for reviewing our work. As many others have previously shown (*e.g.*, Brazauskas & Logan, 2016) regression models with appropriate covariate selection yield results that are comparable to covariate matched-pair analyses. The N3C programming environment is a shared resource with finite memory and processor capacities. A pragmatic decision was made to reduce the control group to a size comparable to the Covid group. We chose a less computationally intensive approach because computer processing performance was severely limited by the very large data. It should be noted that our final sample size was 605,330 unique patients, which was sufficient to discern an association between post-recovery COVID and incident heart failure

Brazauskas & Logan. "Observational Studies: Matching or Regression?" *Biol. Blood Marrow Transplant.* 2016 22(3): 557-563. doi:10.1016/j.bbmt.2015.12.005

2) In their methods, the authors do not provide sufficient information about how incident HF (the primary outcome) was assessed. In their limitations, the authors state 'We were unable to consider HF biomarkers or cardiac imaging...and the majority of HF documentations were non-specific, e.g. 'congestive heart failure'.' So were only administrative diagnosis codes used, or did they have access to laboratory values (i.e., BNP, NT-proBNP)? This information is critical for readers to understand the internal and external validity of the analyses conducted.

We appreciate this opportunity to clarify. The heart failure outcome was based on physician diagnosis and documentation of heart failure in the electronic health record. This is clarified in the Methods. As previously described (Haendel *et al* 2020)*, clinical terms were programmatically extracted from the electronic health records and harmonized to a common data model. Most descriptions from the common data model were either "Heart failure" or "Congestive heart failure", which did not allow us to differentiate heart failure subtypes (*i.e.*, heart failure with preserved vs. reduced ejection fraction)

Biomarkers of cardiac injury (natriuretic peptides and troponins) are now shown in Table 3. It should be noted that laboratory tests were clinically indicated, and were not ordered at the index hospitalization for all patients. Although our definition of heart failure was based on physician diagnosis documented in the EHR, we now include a sensitivity analysis excluding any patients with elevated natriuretic peptides (BNP >100 pg/mL or NT-proBNP >300 pg/mL) at the index hospitalization, as this may suggest existing heart failure at the time of index hospitalization.

After the exclusion of these patients, COVID-19 hospitalization remained associated with post-discharge incident heart failure. This is now reported in the Results.

*Haendel, *et. al.* “The National COVID Cohort Collaborative (N3C): Rationale, design, infrastructure, and deployment”. *JAMIA*. 2020. 28(3): 427-443. doi: 10.1093/jamia/ocaa196

3) Limitations above aside, the authors' analyses as conducted found a significantly higher incidence of HF among patients hospitalized with COVID-19 vs. those hospitalized without COVID-19, even after adjustment for relevant confounders. That is certainly a noteworthy finding, particularly in the large population studied. However, aside from a handful of subgroup analyses, the authors do not include any further in-depth analyses. For example, they don't provide any analyses incorporating troponin I or other markers of cardiac injury. I suspect this may be because they only had access to administrative diagnosis codes and not laboratory data. If they did have access to laboratory data, then I'd like to see some of that data and how it influenced the stated findings. If they didn't have access, then that should be clearly stated in the manuscript.

We agree that biomarkers of cardiac injury are an important consideration. Table 3 now shows available biomarkers of cardiac injury (BNP, NT-proBNP, cardiac troponin T, cardiac troponin I) at the index hospitalization. These laboratory tests were clinically indicated rather than administered as screening tests. Most of the index hospitalizations were for causes other than cardiac conditions. Interestingly, a higher proportion of patients with COVID-19 were tested for biomarkers of cardiac injury. After excluding any patients with elevated BNP or NT proBNP at the index hospitalization, COVID-19 infection remained associated with post-discharge incident heart failure.

4) In their methods, the authors state they excluded 'any patients with history of HF within 2 years prior to the hospitalization.' Why did the authors not choose to exclude patients with ANY history of HF? Why the 2-year cutoff?

The N3C study extracted electronic health records from 2018 onwards. We excluded any patients with historical documentation of HF in the EHR. Some patients with historical documentation of HF may have had HF first diagnosed well before 2018. We have reworded the Methods for clarification.

Reviewer #2 (Remarks to the Author):

1. The novelty is somehow limited as similar data are variably published over the last two years.

We agree that an association between post-recovery COVID-19 infection and incident heart failure has been previously reported. This finding was first published by Xie, *et al.* (*Nature Medicine*, 2022)*, and during this time, our manuscript was under review. It should be noted that *Nature Communications* is “committed to disregard from our editorial evaluation any competing works that are published while a submission to our journal is under review or under revision by

the authors.” The policy is described here: Strength in numbers | Nature Communications . The analysis by Xie *et al* was limited to military veterans, which are predominantly men. We extend this work by showing an association between post-recovery COVID-19 infection and incident heart failure in a nationally representative sample. The novelty of our study population has been clarified in the Introduction and Discussion.

*Xie Y, Xu E, Bowe B and Al-Aly Z. Long-term cardiovascular outcomes of COVID-19. *Nat Med.* 2022;28:583-590

2. Furthermore, the amount of data presented and the analysis performed may not be sufficient for a full paper.

This is a valid point. Our manuscript was originally drafted as a “Brief Communication” (2000 word count limit). The editor has permitted us to expand the manuscript into a full-length article. We have expanded the analysis and presentation accordingly.

3. Please refrain from using specific dates in the introduction.

This comment is appreciated. Our analysis has been updated with hospitalizations and post-hospitalization outcomes up to 3/31/2022. However, we have removed these specific dates from the Introduction.

4. The introduction may be significantly shortened and resentence to improve clarity

Thank you for this suggestion. The Introduction has been revised.

5. Please use the correct manuscript formatting as per journal style

The editor has confirmed that our original formatting for a “Brief Communication” (2000 word count limit) may be expanded to a full length article.

6. The discussion is quite superficial. It does not encompass the complexity of the problem and does not refer to the appropriate literature for discussion.

We certainly agree that the original Discussion section was limited by the “Brief Communication” word count. The revised Discussion has been expanded to a depth appropriate for a full length article.

7. The authors should investigate the causes of the increased incidence of HF as well as the time to hospitalization.

We hypothesize that COVID-19 infection is a cause for incident heart failure. This hypothesis was not refuted by multivariable modeling with adjustment for demographics (age, race, sex), known heart failure risk factors (hypertension, obesity, coronary artery disease, diabetes, chronic

kidney disease, chronic lung disease) or cardiovascular medications at the index hospitalization (angiotensin converting enzyme inhibitor, angiotensin II receptor blocker, beta blocker, and statins). Our hypothesis is also supported by a recent publication analyzing data from the Veteran's Administration Hospitals (Xie, *et al.* 2022, *Nature Medicine*).* We speculate that the pathophysiological link between COVID-19 infection and incident heart failure may be related to inflammation, metabolic changes, endothelial dysfunction, and prothrombotic states. This is detailed in the Discussion.

*Xie Y, Xu E, Bowe B and Al-Aly Z. Long-term cardiovascular outcomes of COVID-19. *Nat Med.* 2022;28:583-590

Was this within 30 days from discharge? Please divide in subgroups accordingly.

Thanks for this suggestion. The average time to heart failure is now shown in Table 2. A similar proportion of incident heart failure cases occurred within 30 days of the index hospital discharge, both for patients with post-recovery COVID (33%), and patients without post-recovery COVID (30%). This is now clarified in the Results.

8. The lack of clinical characteristics is a critical limitation.

We agree that the presentation of details was limited. We have added Table 3, which displays available biomarkers of cardiac injury (BNP, NT-proBNP, cardiac troponin T, cardiac troponin I) at the index hospitalization. These laboratory tests were clinically indicated rather than administered as screening tests.

For the statistical analysis, we additionally adjusted for cardiovascular medications at the index hospitalization (angiotensin converting enzyme inhibitor, angiotensin II receptor blocker, beta blocker, and statins).

9. Table 1 is minimal. It should be significantly expanded with characteristics of patients at discharge and, if possible, with those at first HF event

This comment is appreciated. Table 1 has been expanded to show cardiovascular medication use at the index hospitalization. We have also included Table 3, which presents available biomarkers of cardiac injury (BNP, NT-proBNP, cardiac troponin T, cardiac troponin I) at the index hospitalization.

10. The patients with unknown race/ethnicity should be excluded and not counted as "other"

Thanks for pointing this out. The "Other" category for Race / Ethnicity includes the following: Multi-race, Other race, and Hispanic ethnicity. While Hispanic ethnicity is not a race, it is usually treated as a separate demographic entity from White, Black, or Asian races. Most patients with unknown race were reported as having Hispanic ethnicity. Only 6% of patients

were reported to have unknown race if not Hispanic ethnicity. These have been excluded and annotated in the footnote below Table 1.

11. Please report the % of missing variables

This is an excellent point. The number and percentage of patients with missing data are presented as a footnote below Table 1. For the medical histories, diagnosed conditions were considered present or absent if noted in the electronic health record. This has been clarified in the Methods. Medical histories were queried from all encounters (hospitalizations and outpatient visits) since 2018 up to the index hospitalization.

12. A table with HRs in univariate and multivariate should be reported. However the lack of clinical variables downplays the significance of the analysis as the correction might be underestimated.

We agree that opinions differ regarding modeling approaches. A sequential modeling strategy is now shown in Table 4, which begins with no adjustments, followed by minimal adjustment for demographics (age, race, and sex), followed by additional adjustments for heart failure risk factors (hypertension, obesity, coronary artery disease, diabetes, chronic kidney disease, chronic lung disease), and finally the full model which additionally adjusts for cardiovascular medications at the index-hospitalization (angiotensin converting enzyme inhibitor, angiotensin II receptor blocker, beta blocker, and statins).

In total, our full model adjusts for 13 covariates, which are known to be associated with incident heart failure. A similar analysis from the Veteran's Administration Hospitals adjusted for the following: "age, race, sex and other cardiovascular risk factors, including obesity, hypertension, diabetes, chronic kidney disease and hyperlipidemia (Xie, *et al.* 2022, *Nature Medicine*)

Xie Y, Xu E, Bowe B and Al-Aly Z. Long-term cardiovascular outcomes of COVID-19. *Nat Med.* 2022;28:583-590

REVIEWERS' COMMENTS

Reviewer #1 (Remarks to the Author):

This manuscript by Salah et al. is a revision of a prior manuscript I previously reviewed. I am pleased to see that the authors comprehensively addressed all of my prior comments and concerns. I have no further comments/suggestions for improvement and I congratulate the authors on a job well done!

Reviewer #2 (Remarks to the Author):

The authors replied to all my comments. However, there are some other issues to be addressed.

- please check the correctness of the. number of patients reported and provide a flowchart for exclusion/inclusion.
- this kind of studies cannot ascertain causation. The authors need to significantly downplay the association between COVID and subsequent incidence of HF.

Reviewer #2

1. Please check the correctness of the number of patients reported and provide a flowchart for exclusion/inclusion.

Thank you for this suggestion. We have double checked the number of patients and verified the statistical analysis. Some typos or inaccuracies were noted and have been corrected in the manuscript text (Abstract and Results section), tables (Table 4), and figures (Figure 2). After these corrections, the model estimates and the interpretations were largely unchanged.

We include the study population selection flowchart (Figure 1), and have also provided more details of the study population derivation in the Results section (first paragraph).

2. This kind of studies cannot ascertain causation. The authors need to significantly downplay the association between COVID and subsequent incidence of HF.

We agree with the reviewer. It was not possible for us to randomize patients to have vs. not have COVID-19 hospitalization, and our analysis is purely observational. All model results are clarified in the writing as "associations". We do not state that COVID-19 hospitalization causes incident heart failure. Rather, our conclusion is that COVID-19 hospitalization is associated with incident heart failure.